

# Virulence of entomopathogenic bacteria *Serratia marcescens* against the red palm weevil, *Rhynchophorus ferrugineus* (Olivier)

Baozhu Zhong, Chaojun Lv, Wenlian Li, Chaoxu Li and Tuo Chen

Coconut Research Institute, Chinese Academy of Tropical Agricultural Sciences, Wenchang, Hainan Province, China

Corresponding authors
Baozhu Zhong, baozhuz@163.com
Chaojun Lv, lcj5783@126.com

## ABSTRACT

**Background:** The red palm weevil (RPW), *Rhynchophorus ferrugineus* (Olivier), is an important quarantine pest, which has caused serious economic losses in various palm species, such as coconut, oil palm and date palm. Finding effective biocontrol resources is important for the control of this pest and the protection of palm crops.

**Methods:** A pathogenic strain HJ-01 was isolated from infected and dead pupa of *Tenebrio molitor* using tissue separation method. The HJ-01 strain was streak cultured and purified, and its morphological, physiological, biochemical characteristics, and 16S rDNA homology were identified after conducting a pathogenicity test on RPW larvae.

**Results:** Strain HJ-01 exhibited remarkable pathogenicity against RPW larvae. Under the concentration of HJ-01 suspension was $1.0 \times 10^8$ cfu/mL, the mortality rate of RPW reached 82.22%, and the half-lethal time ($LT_{50}$) was 4.72 days. RPW larvae infected with strain HJ-01 showed reduced movement, decreased appetite, and eventual death. As the treatment progresses, the larvae's bodies turned red, became soft, and started to rot, resulting in the discharge of liquid. HJ-01 demonstrated the ability to produce scarlet pigment after 24 h of culture on a basic medium. Colonies of HJ-01 appeared convex, bright red, moist, and viscous, opaque in the center, irregular at the edges, and emitted an unpleasant odor. Under microscopic observation, the cells of HJ-01 appeared as short rod-shaped and flagellate, with a size ranging from (1.2–1.8) μm × (1.0–1.2) μm. Genomic DNA extraction was performed on the strain, and the 16S rDNA sequence was amplified, yielding a sequence length of 1445 bp. The sequence of strain HJ-01 displayed a 99.72% similarity to that of *Serratia marcescens*. Phylogenetic tree analysis further confirmed that strain HJ-01 belonged to *S. marcescens*. Therefore, the strain HJ-01 has a very good lethal effect on RPW larvae, and it may be used as an effective bacterium for the control of RPW.

## INTRODUCTION

The red palm weevil (RPW), *Rhynchophorus ferrugineus* (Olivier), belonging to the family Curculionidae, is a significant global quarantine pest which native to Southern Asia and

Melanesia (*Nurashikin-Khairuddin et al., 2022*; *EPPO, 2023*). It poses a major threat to various palm species, including *Cocos nucifera*, *Elaeis guinensis*, *Phoenix dactylifera*, *Areca catechu*, and other ornamental palms (*Wang et al., 2013*; *Lü et al., 2020*). The RPW primarily causes damage through larval burrowing, characterized by its destructive nature, high lethality, and difficulty in early detection. In the Middle East, the annual economic losses due to the RPW are estimated to range from $5–25 million, with Saudi Arabia alone accounting for $1.74–8.69 million (*Massoud et al., 2012*). In China, the RPW was first reported in Zhongshan, Guangdong Province (*Wan, Zheng & Guo, 2005*), and has since spread to 15 provinces and cities, causing severe damage to palm plants in Guangdong, Hainan, Yunnan, and other regions (*Han et al., 2013*). Infestations by RPW significantly weaken the palm trunks, reducing their productivity and compromising their ability to withstand environmental conditions, such as strong winds (*Saleh, 2018*). Currently, the primary methods of control involve the use of chemical insecticides (*Reyad et al., 2020*; *Milosavljević et al., 2022*), pheromone trapping ( *El-Shafie & Faleiro, 2017*; *Dalbon et al., 2021*; *Al Ansi et al., 2022*) and fumigants (*Llácer & Jacas, 2010*; *Wakil et al., 2018*). Research on biological control mainly focuses on the use of natural enemies (*Löhr, Negrisoli & Molina, 2019*), entomopathogenic nematodes (*El Sadawy et al., 2020*; *Rehman & Mamoon-ur-Rashid, 2022*), entomopathogenic bacteria (*Yasin et al., 2021*), and entomopathogenic fungi (*Al-Keridis, Gaber & Aldawood, 2020*; *Ziedan et al., 2022*).

*Serratia marcescens* (Enterobacteriales: Enterobacteriaceae), also known as Bacillus spiritus, belongs to Serratia of enterobacteriaceae and is a gram-negative bacterium, which is widely existing in nature (*Petersen & Tisa, 2013*). During the growth process, this bacterium can produce a secondary metabolite, linomycin, which has attracted increasing attention worldwide due to its highly pathogenic to a variety of agricultural and forestry pests (*Fu et al., 2019*; *Hu et al., 2021*; *Tao et al., 2022*). In this study, a bacterium was isolated from the dead pupae of mealworm, *Tenebrio molitor* (Coleoptera: Tenebrionidae). After purification and back splicing tests on larvae and pupae of *T. molitor*, it was identified as an insect pathogenic bacterium, named HJ-01. The morphological characteristics, physiological and biochemical properties of the bacterium were observed, and the 16S rDNA of the strain was extracted for homology analysis. The strain was ultimately identified as *S. marcescens*. Its pathogenicity against RPW larvae was tested to explore its potential for biological control. The findings aim to provide valuable insights for the selection of biological control resources and the development of biological control technologies for RPW.

## MATERIALS AND METHODS

### Isolation and culture of the bacterial strain

Naturally infected and deceased pupae of *T. molitor* were collected from Wenchang, Hainan Province, China. The samples underwent a series of steps for preparation according to tissue separation method described by *Fang (1998)*. Firstly, the samples immersed in 70% alcohol for 1 min and then rinsed with sterile distilled water. Next, the samples were surface-sterilized using 0.1% mercury chloride and washed three times with sterile distilled water. Subsequently, sections of the tissues were cut and inoculated onto

Luria-Bertani solid medium (LB), which consisted of 10 g/L peptone, 5 g/L yeast, 5 g/L sodium chloride, and 15 g/L agar. The inoculated tissues were placed on separate sterile petri dishes, sealed with Parafilm, and incubated at 28 ± 1 °C with a relative humidity of 75 ± 5% and photoperiod (L:D) of 8:16 for 24 h. A single colony exhibiting red pigment production was selected and cultured on LB solid medium for purification. To confirm the strain's ability to produce red pigment, a backgrafting test was conducted by introducing the bacterial solution to a healthy *T. molitor* specimen. This process aimed to restore the strain capable of producing red pigment, which was designated as HJ-01.

## Pathogenicity determination of strain HJ-01 against RPW

The RPW was provided by the Biological Control Laboratory of Coconut Research Institute of Chinese Academy of Tropical Agricultural Sciences (CRI-CATAS) and was bred for more than three consecutive generations on semi-artificial diets (*Ma et al., 2012*), and the larvae with same age and the same size were selected for the test.

The purified strain was prepared in sterile distilled water containing aqueous 0.05% Tween-80, and the mixture was vortexed to attain homogenization. A dilution series of bacterial suspension ($1.0 \times 10^8$, $1.0 \times 10^7$, $1.0 \times 10^6$, $1.0 \times 10^5$, $1.0 \times 10^4$ cfu/mL) was prepared thorough mixing, then sprayed on larvae. Larvae sprayed with distilled water served as control. Then the larvae were transferred to the artificial feed cups for further incubation, 1 larva per cup, 20 larvae per treatment, replicated 3 times. The larvae were kept in controlled conditions (28 ± 1 °C, 75 ± 5%RH and a photoperiod of 8:16 h L:D) and checked daily for mortality. The dead larvae were reisolated using moisturizing the culture and verifying the pathogenicity of their isolates according to Koch's rule.

## Morphological, physiological and biochemical identification of strain HJ-01

The morphology was observed using an optical microscope, and the physiological and biochemical reaction tests were identified by reference to methods such as bacterial classification and systematic identification (*Dong & Cai, 2001*).

## 16S rDNA amplification and sequence analysis of strain HJ-01

### Genomic DNA extraction

The purified strain was inoculated in a triangular flask with sterilized LB liquid medium and incubated on a shaker at 28 °C and shaken at 180 rpm for 24 h. The genomic DNA of strain HJ-01 was extracted by modified CTAB method (*Zhu, Chen & Chen, 2013*).

### Amplification and Determination of 16S rDNA Sequence

The 16S rDNA sequences of strain HJ-01 were amplified by PCR using universal primer sets 27F (5′- AGAGTTTGATCCTGGCTCAG-3′) and 1492R (5′-GGTTACCTTGTTACGACTT-3′) (*Zhu, Chen & Chen, 2013*). The PCR reactions (50 μL) contained: 25 μL of 2× Taq PCR premix reagent, 1 μL each of primers 27F and 1492R at 20 μmol/L; 2 μL of template DNA; 21 μL of double-distilled water. The PCR were conducted under the following conditions: at 94 °C for 5 min, 35 cycles at 94 °C for 30 s, 58 °C for 30 s, and 72 °C for 90 s, followed by a final elongation at 72 °C for 10 min. PCR
products were kept at 4 °C. The size and quality of PCR products were detected by 1% agarose gel electrophoresis. Then the remaining PCR products were sequenced by Sangon Biotech Co., Ltd. (Shanghai, China).

### Construction of phylogenetic tree for strains

The sequences obtained were analyzed against nucleic acid data in GenBank using NCBI's BLAST tool, 16S rDNA sequences of related strains were downloaded, homology analysis was performed using multiple sequence alignment with MEGA V.6.0 and phylogenetic tree was constructed. The conformation and stability of the phylogenetic tree was determined by sampling and analysis 1,000 times with MEGA V.6.0 software.

## Statistical analysis

The mortality rates were analyzed using a one-way analysis of variance. Abbott's formula (Abbott, 1925) was applied to correct the mortality percentage if the control mortality was between 5 and 20%. The lethal time value ($LT_{50}$) and the corresponding 95% fiducial limits were calculated by using SPSS V. 21.

## RESULTS

## Isolation of strains and pathogenicity to RPW larvae

A red pigment-producing strain was isolated from infected *T. molitor* pupae (Fig. 1A), which was purified on LB medium and then tested against mealworm pupae, allowing the strain to be isolated again and named HJ-01 (Fig. 1B).

Strain HJ-01 was incubated in a constant temperature shaker for 24 h and then inoculated with RPW larvae to observe their infection status (Figs. 1C–1F). After 8 h, the larvae displayed reduced mobility and loss of appetite. The larvae started to show signs of death after treated 24 h and the number of larval deaths gradually increased with the extension of infection. The infected larvae began to turn redden and soften at 48 h after death. As time progressed, the larvae further deteriorated, with their bodies decaying and releasing red liquid. The HJ-01 strain was isolated from the carcasses of the infected larvae, so it can be determined that the HJ-01 strain is the pathogen that caused the death of RPW.

Strain HJ-01 exhibits remarkable pathogenicity towards RPW larvae. Regardless of the concentration tested, the strain effectively eliminates the larvae, and the mortality rate increases with longer treatment duration (Fig. 2). The highest larval mortality rates were observed at suspension concentrations of $1.0 \times 10^8$ cfu/mL and $1.0 \times 10^7$ cfu/mL, reaching cumulative mortality rates of 82.22% and 77.78%, respectively. The $LT_{50}$ of RPW larvae caused by strain HJ-01 was 4.72 days at concentration of $1.0 \times 10^8$ cfu/mL (Table 1, $p < 0.05$).

## Morphological observation, physiological and biochemical characteristics of strain HJ-01

The strain was cultured on LB solid medium for 24 h and began to produce red pigment, the colony was raised, bright red, moist and sticky, opaque in the center, irregular at the

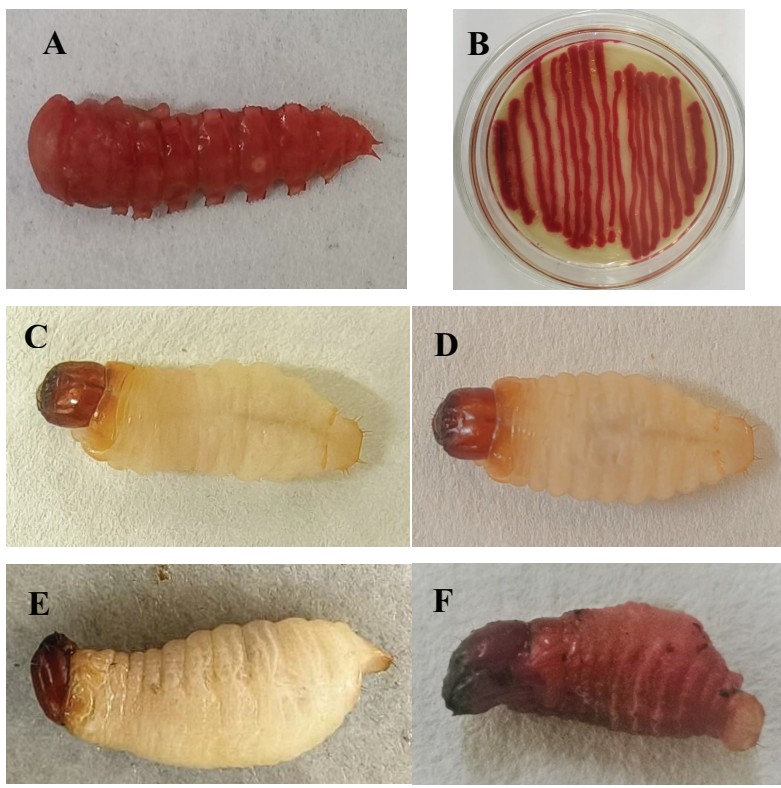

**Figure 1 *Serratia marcescens* strain HJ-01 and infected insects.** (A) Diseased *Tenebrio molitor* pupae. (B) Strain HJ-01 of *Serratia marcescens*. (C) *Rhynchophorus ferrugineus* larvae before infection. (D) The larvae displayed reduced mobility and loss of appetite after 8 h infected with *S. marcescens*. (E) The larvae was died after 24 h infected with *S. marcescens*. (F) Infected larvae turned red and soft at 48 h after death.

edges, and smelly. Under the microscope, the bacterium was short rod-shaped, flagellated, and the size was (1.2–1.8) μm × (1.0–1.2) μm.

Table 2 presents the results indicating that strain HJ-01 is Gram-negative and facultative aerobic. It exhibited positive reactions for the Voges-Proskauer (V-P) test, motility test, glucose acid production, and gas production. However, it showed negative results for the methyl red test and phenylpropyl amino acid decarboxylase reaction. In terms of carbohydrate utilization, the strain produced acid when grown on media containing sucrose, maltose, sorbitol, and mannitol, while it did not produce acid when grown on media containing lactose, raffinose, fibrinous disaccharide, xylose and arabinose. These physiological and biochemical characteristics, as determined through standard methods outlined in the Manual of Systematic Identification of Common Bacteria and Bergey's Manual of Determinative Bacteriology, confirm that this strain belongs to *S. marcescens*.

## Amplification and analysis of 16S rDNA sequence of strain HJ-01

The PCR amplification product of strain HJ-01 was analyzed using 1% agarose gel electrophoresis, revealing a distinctive band of approximately 1,400 bp in size (Fig. 3A). The amplification product was subsequently sent to Sangon Biotech Co., Ltd (Shanghai,

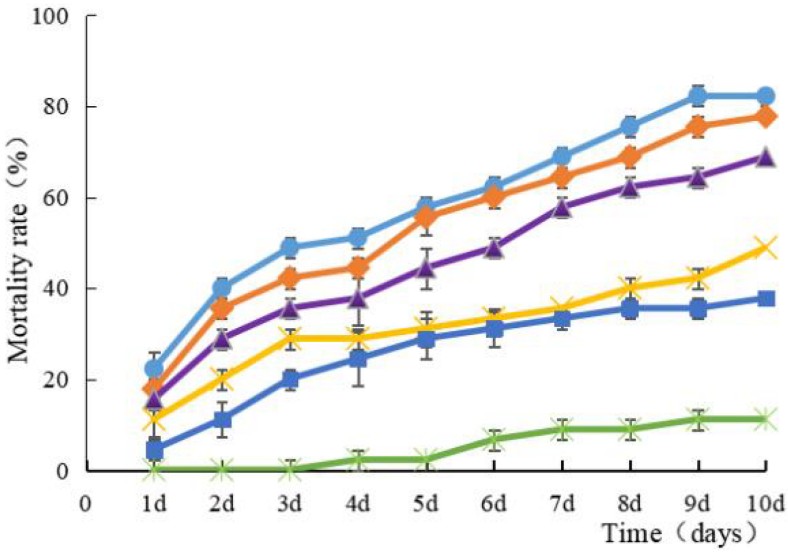

**Figure 2 Mean mortality (% ± SE) of *Rhynchophorus ferrugineus* larvae after various exposure intervals infected with *S. marcescens*.** Concentration of bacterial suspension cfu/mL: filled circle 1 × $10^8$, filled diamond 1 × $10^7$, filled triangle 1 × $10^6$, filled cross 1 × $10^5$, filled square 1 × $10^4$, filled rice characters CK; The error bars in the figure indicate the standard error (SE) of three repetitions.

**Table 1 LT$_{50}$ values of *S. marcescens* strain HJ-01 tested against larvae of *Rhynchophorus ferrugineus*.**

| Bacterial suspension (cfu/mL) | LT$_{50}$ (days) | Correlation coefficient r | 95% Confidence interval | |
| --- | --- | --- | --- | --- |
| | | | Lower | Upper |
| $1.0 \times 10^8$ | 4.72 | 0.9845 | 4.24 | 5.26 |
| $1.0 \times 10^7$ | 5.30 | 0.9932 | 4.70 | 5.98 |
| $1.0 \times 10^6$ | 6.83 | 0.9950 | 5.90 | 7.92 |
| $1.0 \times 10^5$ | 14.81 | 0.9632 | 9.43 | 23.26 |
| $1.0 \times 10^4$ | 22.66 | 0.9867 | 9.08 | 56.58 |

**Note:**
LT$_{50}$, lethal time for 50 % mortality.

China) for sequencing, resulting in a full-length sequence of 1,445 bp (Fig. 3B). The obtained sequence was uploaded to GenBank, and its accession number is OP317557. By performing a BLAST search in the NCBI nucleic acid database, it was found that the 16S rDNA nucleotide sequence of strain HJ-01 exhibited a high similarity to that of *S. marcescens* strain whpu-5 (accession number: MK157269.1), with a sequence similarity of 99.72%. These findings suggest that strain HJ-01 is likely to be *S. marcescens*.

## Phylogenetic tree of Strain HJ-01

A total of 12 closely related strains belonging to the Serratia genus were selected from the nucleic acid database for multiple sequence alignment with the 16S sequences of HJ-01. The aligned sequences were then used to construct a phylogenetic tree using MEGA V.6.0 software, employing the neighbor-joining (NJ) method with a bootstrap value of 1,000. The phylogenetic analysis revealed that strain HJ-01 shared the highest similarity with

**Table 2 Physiological and biochemical characteristics of strain HJ-01.**

| Characteristics | *Serratia marcescens* | HJ-01 | Characteristics | *Serratia marcescens* | HJ-01 |
|---|---|---|---|---|---|
| Gram staining reaction | − | − | Maltose | + | + |
| Methyl Red | − | − | Sucrose | + | + |
| V-P | + | + | Lactose | − | − |
| Movement test | + | + | Raffinose | − | − |
| Glucose acid production | + | + | Fibrinose | − | − |
| Glucose gas production | + | + | D-xylose | − | − |
| Phenylpropyl amino acid decarboxylase | − | − | Arabinose | − | − |
| D-Mannitol | + | + | D-sorbitol | + | + |

**Note:**
"+" is positive; "−" indicates negative.

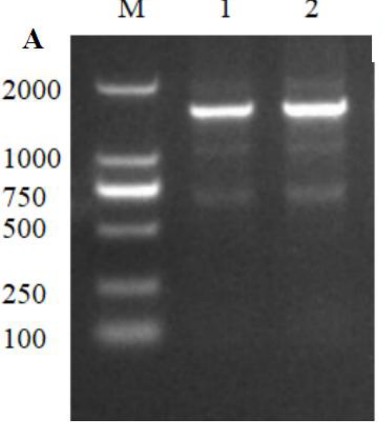

**Figure 3 Electrophoresis and nucleotide sequence of 16S rDNA PCR products of strain HJ-01.** (A) Electrophoresis of 16S rDNA PCR products of strain HJ-01, M: DL 2,000 marker. 1 and 2: product of 16S rDNA. (B) Nucleotide sequence of 16S rDNA of strain HJ-01.

*S. marcescens* strains, specifically with accession numbers MK157269.1 and AB680122.1, exhibiting a self-extension value of 90% (Fig. 4).

In conclusion, based on the morphological characteristics, physiological and biochemical traits, as well as the identification results of 16S rDNA, it has been established that strain HJ-01 belongs to the species *S. marcescens* within the genus Serratia.

## DISCUSSIONS

Biological control refers to the utilization of organisms, microorganisms, and their byproducts to manage pests. It is an essential component of IPM, offering a safe and environmentally friendly approach. Therefore, the discovery of safe and effective biological control resources is of utmost importance (*Roberts et al., 2007*). Among the bacteria suitable for pest control, Serratia is widely distributed in nature and can be isolated from healthy, infected, or deceased insects. Among the Serratia genus, *S. plymuthica* and *S. entomophila* have been extensively studied. *S. plymuthica* HRO-C48, registered and commercialized in Germany under the trade name Rhizostar is primarily used to combat

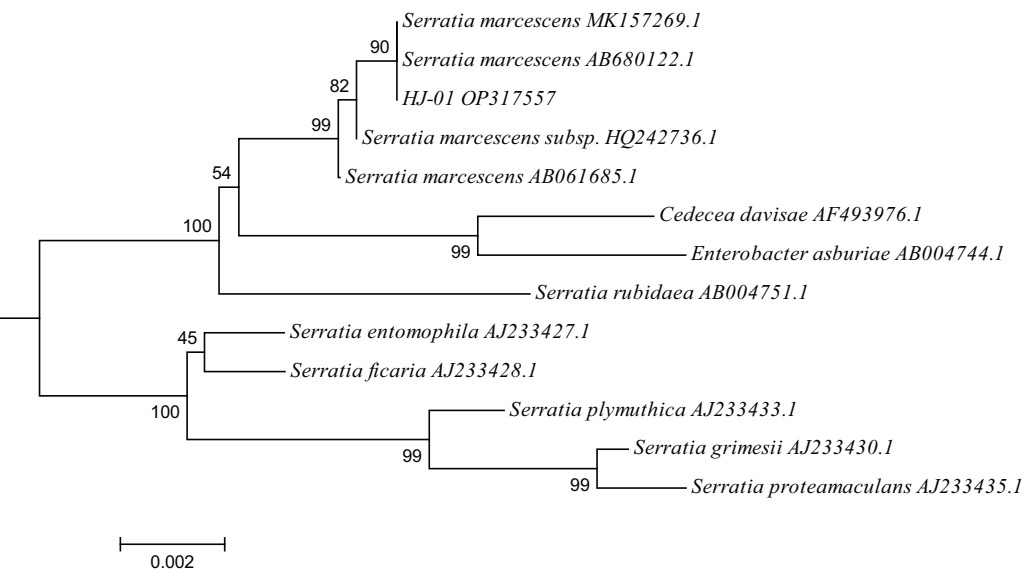

**Figure 4 Phylogenetic placement of strain HJ-01 based on 16S rDNA.** The aligned sequences of 12 closely related strains belonging to the Serratia genus were selected from the nucleic acid database and then were used to construct a phylogenetic tree using MEGA V.6.0 software, employing the neighbor-joining (NJ) method with a bootstrap value of 1,000. The scale bar represents a genetic variability of 0.002 for the genome.

root rot and wilt in strawberry plants (*Berg, 2009*). On the other hand, *S. entomophila* is predominantly employed for the biological control of scarab beetles (*Nuñez-Valdez et al., 2008*).

*S. marcescens* can produce a variety of exoenzymes, including chitinase, which can hydrolyse and destroy the surface and periplasmic structure of insects, causing insect death eventually (*Zhao et al., 2020*; *Chin et al., 2021*). Studies have found that *S. marcescens* has pathogenicity against a variety of Coleoptera insects. *Yang et al. (2014)* isolated a strain of *S. marcescens* PS-1 from diseased *Phyllotreta striolata* (Coleoptera: Chysomelidae) larvae, which was highly pathogenic to *P. striolata* adults. *Deng et al. (2008a, 2008b)* isolated a strain of *S. marcescens* from the carved grooves of *Anoplophora glabripennis* (Motschulsky) (Coleoptera: Cerambycidae). The fatality rate reached 80.6% at $7.8 \times 10^{10}$ cfu/mL after applied the bacterial solution to the larvae using a microinjector. *Zhang et al. (2011)* isolated *S. marcescens* subspecies HN-1 from eggs and dead larvae of RPW, and using this bacterium to infect larvae resulted in a 60% mortality and an 80% reduction in egg hatching rate. In this study, a red pigment-producing strain HJ-01 was isolated from the dead mealworm pupae, and the morphological characteristics, physiological and biochemical characteristics were identified to be consistent with those of *S. marcescens*. By extracting 16S rDNA sequence of the strain, the similarity of 16S rDNA sequence between HJ-01 and *S. marcescens* was 99.72%. Therefore, the strain HJ-01 could be identified as *S. marcescens*.

The pathogenicity of *S. marcescens* strains varies according to its source, application method and pest species. *S. marcescens* isolated from *Helicoverpa armigera* (Hübner) (Lepidoptera: Noctuidae) by *Bulla, Rhodes & Julian (1975)* has pathogenicity against not

only *H. armigera*, but also *Pieris rapae*, a member of the family Pieridae. However, it was less pathogenic to the larvae of *Spodoptera exigua* Hübner (Lepidoptera: Noctuidae), which belongings to the same family. The strain TC-1 of *S. marcescens* isolated from naturally infected *Anomala corpulenta* (Coleoptera: Rutelidae) has larvicidal activity against *Plutella xylostella* (Lepidoptera: Plutellidae), *S. exigua*, *H. armigera*, *Bombyx mori* (Lepidoptera: Bombycidae) and the nematode *Caenorhabditis elegans* (*Tao et al., 2022*). *Zhang, Ma & Ma (2020)* and *Zhang et al. (2021)* reported that the lifespan of *Curculio dieckmanni* Faust (Coleoptera: Curculionidae) was shortened significantly when adult were fed on hazelnut leaves containing $1.8 \times 10^8$ cfu/mL *S. marcescens*. In this study, strain HJ-01 of *S. marcescens* isolated from mealworm pupae had a fatality rate of 82.22% against RPW larvae, higher than that of treated with *Bacillus thuringiensis*, which mortality was 46.86–58.36% (*Yasin et al., 2021*), and almost equal to the mortality of RPW larvae treated with commercial bacteria-based biopesticide at a concentration of 2.0 mg/mL (*de Altube & Peña, 2009*). It can be seen that the *S. marcescens* HJ-01 strain has a good lethal effect on RPW larvae, and it may be used as an effective bacterium for the control of RPW. Further studies are required to test the pathogenicity of *S. marcescens* for RPW at different ages and stages, leading to the development of exploitable biological agents and field applications.

## CONCLUSIONS

A strain named HJ-01, exhibiting insecticidal properties against the RPW, was isolated from infected mealworms. Through a comprehensive analysis of its physiological, biochemical, and molecular characteristics, it was identified as *S. marcescens* HJ-01. Upon infection with this strain, RPW larvae displayed reduced activity, a softer texture, and eventually succumbed to the treatment. Notably, the deceased insects emitted red pus upon gentle contact. The concentration of the HJ-01 suspension used in the experiments was $1.0 \times 10^8$ cfu/mL, resulting in an impressive mortality rate of 82.22% and the $LT_{50}$ value for RPW larvae was determined to be 4.72 days. Strain HJ-01 has the potential to be used for biological control of RPW.

### Funding
This study was supported by the Hainan Provincial Natural Science Foundation of China (Grant number: 318MS105) and the Major Planned Science and Technology Project of Hainan Province, P.R. China (Grant number: ZDXM 20120029). The funders had no role in study design, data collection and analysis, decision to publish, or preparation of the manuscript.

### Grant Disclosures
The following grant information was disclosed by the authors:
Natural Science Foundation of China: 318MS105.
Major Planned Science and Technology: ZDXM 20120029.

## Competing Interests

The authors declare there are no competing interests.

## Author Contributions

- Baozhu Zhong conceived and designed the experiments, performed the experiments, analyzed the data, prepared figures and/or tables, authored or reviewed drafts of the article, and approved the final draft.
- Chaojun Lv conceived and designed the experiments, performed the experiments, authored or reviewed drafts of the article, and approved the final draft.
- Wenlian Li performed the experiments, authored or reviewed drafts of the article, and approved the final draft.
- Chaoxu Li performed the experiments, authored or reviewed drafts of the article, and approved the final draft.
- Tuo Chen performed the experiments, authored or reviewed drafts of the article, and approved the final draft.

## DNA Deposition

The following information was supplied regarding the deposition of DNA sequences:
The isolate HJ-01 sequence is available at Genbank: OP317557.

## Data Availability

The raw data is available in the Supplemental File.

## Supplemental Information

Supplemental information for this article can be found online at http://dx.doi.org/10.7717/peerj.16528#supplemental-information.

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
