# Peer review of "Virulence of entomopathogenic bacteria Serratia marcescens against the red palm weevil, Rhynchophorus ferrugineus (Olivier)"

_PeerJ, doi:10.7717/peerj.16528_

## Round 0.1 · original submission · Minor Revisions

Dear authors.
I have reviewed your manuscript, " Isolation and identification of a pathogenic strain of Serratia marcescens against the red palm weevil Rhynchophorus ferrugineus Olivier" with great interest, and I believe it holds significant potential for publication in our journal. However, to ensure the manuscript aligns with our publication standards, there are a couple of crucial points that require attention.
Your manuscript requires a thorough grammar check. It's important that your work is presented in clear and grammatically correct English. Consider engaging a professional proofreading service or a proficient English speaker to enhance the language quality.

Ensure your manuscript adheres to PeerJ's formatting guidelines, covering aspects such as headings, citations, references, figures, and tables. Detailed formatting instructions can be found in the "Author Guidelines" on the PeerJ website.

To expedite the review process and improve your manuscript's overall quality, please address these issues. After making the necessary revisions, resubmit your revised manuscript via our online submission system. Include a response letter detailing the changes made and your reasoning behind them.

**Language Note:** The Academic Editor has identified that the English language must be improved. PeerJ can provide language editing services - please contact us at copyediting@peerj.com for pricing (be sure to provide your manuscript number and title). Alternatively, you should make your own arrangements to improve the language quality and provide details in your response letter. – PeerJ Staff

Reviewer 1 ·

Basic reporting

Zhong and co-authors have summarized the identification and molecular characteristics of Serratia marcescens. They have also observed antagonistic activity of isolate against red palm weevil a notorious pathogen of palm species including coconut and date palm. However, the morphological, physiological, biochemical characteristics were conduct for isolate too. The data are well represented, figures and tables are depicted in well mannered. Importantly, the chemical control for red palm weevil is a conventional technique but the biocontrol an environment friendly and a rare method to cope such devastating pathogens to palm species. The study is significant and may be important in field of agriculture and pest management. The current version of manuscript is nicely represented except for some English/grammatical errors that may be corrected by authors or third party.
Line # 20-21 Please complete the sentence, i.e., controlling and preventing the insect against which crops? I suggest revise the sentence with complete sense.
Line # 23-24. Please revise the sentence.
Line # 29-30, again there is no sense in sentence for larvae of which? I suggest the revision of sentence with complete sense.
Line # 33. They displayed opacity? What do you mean by they?
Line # 246-47. Revise the sentence.
Please format in italic all scientific names for bacteria or red palm weevil. Check in entire text in manuscript.
In discussion section authors have cited several references related to their study. I suggest comparing your results with other conducted work similar work. See whether your results are significant or similar.

Experimental design

well represented.

Validity of the findings

No comments

Reviewer 2 ·

Basic reporting

no comment

Experimental design

no comment

Validity of the findings

no comment

Additional comments

Original Article: Recommendation, Minor Revision
Overview and general recommendation: Biological control bacteria have always been a key focus of disease and pest control, and this paper introduces the pathogenic strain, Serratia marcescens, that could against the red palm weevil Rhynchophorus ferrugineus Olivier. Rhynchophorus ferrugineus is an invasive insect in China, and this study may have significant implications for subsequent biological pest control. The manuscript is well written in some degree, and if authors modified those following comments carefully, I think it can be accepted and published.
Major comments: The author should show images before and after insect infection (Figure 1) for easy comparison and reader recognition. Besides, the author should indicate how long after R. ferrugineus are infected with bacteria, will they die suddenly or gradually? Are insects raised separately and inoculated with bacteria? If a group of insects are raised together and inoculated with bacteria, is there an infection? Have insects been disinfected before being inoculated with bacteria? The author should explain these issues clearly. Finally, in the abstract, I suggest that the author reflect the significant mortality rate of this bacterium, which can highlight its importance as a biological control bacterium.
Minor comments: Overall, many details in the paper need to be carefully corrected by the author, such as word errors (line 19…), missing commas (title, line 19…), missing space symbols (line 20, 41, 49…), abbreviation (line 24, 82, 85…), genus name should be italicized (line 208, 218, 226…), and so on. In addition, for grammar, it is recommended that the author can further improve.

Reviewer 3 ·

Basic reporting

The red palm weevil is the most important pest on palm, which brings serious impact on the growth of palm. Till now,there are still no particularly efficient methods for the control of this insect, and biological control should be emphasized as one of the eco-friendly methods among different control methods. The authors studied the control potential of Serratia marcescens on red palm weevil, which is a very important reference value for the research of biological control technology of red palm weevil. However, there are still parts of the manuscript that need to be upgraded, and if these can be revised and supplemented, I think it can be accepted and published.

Experimental design

The resistance of different insect populations to pathogens is different. Is the red palm weevil used in manuscript collected indoors or in the field? If it is raised indoors, what materials are used as food? If it is collected in the field, what host is it collected from? These should be necessary in section “Materials and methods”.

Validity of the findings

The resistance of different insect populations to pathogens is different. Is the red palm weevil used in manuscript collected indoors or in the field? If it is raised indoors, what materials are used as food? If it is collected in the field, what host is it collected from? These should be necessary in section “Materials and methods”.
The infection of biocontrol bacteria to pests is a gradual process, and only a photo after infection is provided in this paper. Therefore, it is necessary to provide photos of different infection periods to illustrate the infection process. In the morphological effects of S. marcescens on red palm weevil, there is a lack of effective control pictures, which need to be supplemented;
Since the purpose of this study is ultimately to use Serratia marcescens for the prevention and treatment of red palm weevil, the potential of S. marcescens for the prevention and treatment of red palm weevil and which studies need to be further developed should be described in the discussion or conclusion section.

Additional comments

Overall, there are some details that need attention, such as line86, ' Hainan Island ' should be ' Hainan province ' ; line145, ' MEGA6.0 ' should be ' MEGA V.6.0 ' ; line 240, ' red palm weevil ( RPW ) '. Since ' red palm weevil ' appears in the preface of the article, the abbreviation should be mentioned early. It is hoped that the author will read the article carefully and correct these details.

Reviewer 4 ·

Basic reporting

The MS titled “Isolation and identification of a pathogenic strain of Serratia marcescens against the red palm weevil Rhynchophorus ferrugineus Olivier” is submitted to Peer J. I went through the whole MS and found that this article is worth publishing in this journal with major changes. Attached below are my comments:
1. Suggested edits are marked as tack changes.
2. The MS is written well, but with a few grammatical mistakes. Consult with some fluent English-speaker to improve the write-ups.
3. The title needs to be rephrased to look more attractive.
4. Consult more recent literature, as most of the literature is old.
5. The Materials and Methods section must be supported with strong references.
6. No statistical analysis has been provided for mortality or LT50 calculation.
7. Table and figure legends must be rephrased.
8. Cross-check the references carefully and format them according to the journal guidelines.

Experimental design

The study is well designed and organized but needs statistical analysis for mortality and LT50 in materials and methods section

Validity of the findings

The findings are worth considering and provide a foundation for effective biological control of RPW using microbial control agents, particularly Entomopathogenic bacteria.

Additional comments

The MS can be approved for publication after the incorporation of suggested inputs. Authors are advised to refine the document.

Annotated reviews are not available for download in order to protect the identity of reviewers who chose to remain anonymous.

---

## Round 0.2 · accepted · Accept

The authors of this manuscript addressed all concerns positively, therefore, my recommendation is to accept this manuscript.

Reviewer 2 ·

Basic reporting

Revised manuscript: qualified language, qualified references, and professional figures and tables.

Experimental design

Revised manuscript: scientific issues have certain significance, performed to a high technical & ethical standard, and have reasonable description methods.

Validity of the findings

Revised manuscript: with a certain degree of innovation, data statistics are correct, and conclusions are linked to original research question & limited to supporting results.

Additional comments

The author has carefully revised the paper according to the advice, so with the consent of the editor, I believe that this paper meets the requirements for publication.

Reviewer 3 ·

Basic reporting

no comment

Experimental design

no comment

Validity of the findings

no comment

Reviewer 4 ·

Basic reporting

The MS titled “Isolation and identification of a pathogenic strain of Serratia marcescens against the red palm weevil Rhynchophorus ferrugineus Olivier” is submitted to Peer J. I went through the whole MS after revision and found that the authors have addressed the curies and comments which has improved the MS significantly.

Experimental design

The study is well designed and organized, and statistical analysis has performed well .

Validity of the findings

The findings are worth considering and provide a foundation for effective biological control of RPW using microbial control agents, particularly Entomopathogenic bacteria.

Additional comments

The MS needs to be approved for publication as comments and suggestions have been addressed and incorporated in the MS.